# Comparison of Mechanical Properties of a Self-Adhesive Composite Cement and a Heated Composite Material

**DOI:** 10.3390/polym14132686

**Published:** 2022-06-30

**Authors:** Anastazja Skapska, Zenon Komorek, Mariusz Cierech, Elzbieta Mierzwinska-Nastalska

**Affiliations:** 1Department of Prosthodontics, Medical University of Warsaw, 02-091 Warsaw, Poland; mcierech@wum.edu.pl (M.C.); emierzwinska@wum.edu.pl (E.M.-N.); 2Department of Advanced Technologies, Military University of Technology, 00-908 Warsaw, Poland; zkomorek@wp.pl

**Keywords:** composite materials, preheated composite material, adhesive cementation

## Abstract

(1) Background: Due to the limitations of composite cements, the authors carried out tests to compare such materials with preheated composite materials because the latter may be an alternative to cements in the adhesive cementation procedure. (2) Methods: The materials used in the adhesive cementation procedure, i.e., Enamel Plus Hri (Micerium, Avegno, Italy), a heated composite material, and RelyX U200 Automix (3M, Maplewood, MN, USA), a dual composite cement, were tested for microhardness, compressive strength, flexural strength, diametral compressive strength, and elastic modulus. Composite material was heated to the temperature of 50 degrees Celsius before polymerisation. (3) Results: Higher values of microhardness (by 67.36%), compressive strength (by 41.84%), elastic modulus (by 17.75%), flexural strength (by 36.03%), and diametral compressive strength (by 45.52%) were obtained using the Enamel Plus Hri composite material compared to the RelyX U200 self-adhesive cement. The survey results revealed statistically significant differences. (4) Conclusions: Due to its better mechanical properties, the heated composite material (Enamel Plus Hri) is a beneficial alternative to composite cements in the indirect restoration placement procedure. As the strength parameters of the heated composite material increase, a higher resistance to the compressive and bending forces present in the oral cavity, and hence a greater durability of the created prosthetic reconstructions can be expected.

## 1. Introduction

Due to the principle of maximum possible conservation of hard tooth substance preparations which applies in dentistry, indirect reconstructions, such as veneers, inlay, onlay, or overlay fillings are of increasing interest. Due to the continually improved ceramic materials and the adhesive method of placing such restorations, indirect reconstructions have become an alternative to the conventional treatment of a cavity using crown-root inlays and crowns in many cases [1]. Indirect reconstructions are also used increasingly over large direct restorations. Direct restorations host a range of disadvantages such as polymerisation shrinkage, which results in a marginal gap, low abrasion resistance, colour instability, and risk of uncontrolled fracture of cavity walls.

The indirect restoration cementation procedure is crucial for the long-term success of prosthetic treatment. It relies on creating a strong and permanent connection in two configurations: tooth–cement and cement–reconstruction material. The bond strength depends on the quality of the created micromechanical and chemical connection [2]. Dual composite cements have been successfully applied for this purpose for many years. They are composed of an organic matrix, a powdered ceramic (aluminium-boron-bar glass), and a silane-based bonding agent. The proportions of the organic and inorganic phases are decisive for the achieved strength of the connection between a prosthetic structure and enamel & dentine [3]. The size of the filler particles ranges from 0.04 to 5.0 µm and accounts for 30–75% of the material volume. In order to achieve the proper viscosity, in turn, DEGMA (diethylene glycol dimethacrylate) and TEGDMA (triethylene glycol dimethacrylate) monomers are added to the matrix. A polymer matrix is typically composed of Bis-GMA (bisphenol A-glycidyl methacrylate) in 75% and TEGDMA in 25%. In composite cements, the content of the filler is lower than in composite materials applied for reconstructing hard tooth substances. This results in a higher castability, which enables the creation of a thin layer without the need of excessive force, while deteriorating the mechanical properties of the composite and increasing the polymerisation shrinkage may be responsible for bacterial microleakage [4]. The dual bonding method enables the polymerisation process but with a limited access of light. Another drawback is the often-complex cementation procedure, which requires absolute isolation of the operating field from moisture, which results in longer work and greater sensitivity to mistakes. Despite these inconveniences, composite cements create a stronger connection with tooth substances and ceramic materials than the available conventional cements, such as phosphate, polycarboxylate, glass-ionomer, and resin-modified glass-ionomer cements [5].

The decreased mechanical strength and the increased polymerisation shrinkage encouraged clinicians to attempt to reduce the material viscosity without reducing the quantity of filler. The use of the thermal method to reduce the viscosity of the composite material, used only for filling cavities to date, enabled the application of this material in the adhesive cementation procedure for indirect restorations [1]. Because of the heating procedure, the material acquires new properties. The speed of the monomer conversion increases considerably following a temperature rise in the composite resin [6]. A greater quantity of filler compared to fluid dual cements results in decreased polymerisation shrinkage and, as a consequence, a smaller marginal gap [7]. Tests prove that restorations cemented in this manner are characterised by less excess after polymerisation due to the absence of a chemical catalyst, and hence a longer time of clinical work [8]. Heating a composite material increases its mechanical strength, compressive strength, diametral compressive strength, flexural strength, elastic modulus, microhardness, degree of conversion, and fluidity, while decreasing abrasive wear [5,7,9]. 

The aim of this study is to compare selected mechanical properties of a self-adhesive composite cement to a heated composite material applied in the adhesive cementation procedure.

## 2. Materials and Methods

### 2.1. Materials

The tests were performed using Enamel Plus Hri (Micerium, Avegno, Italy), a nanohybrid composite material with the dentine colour UD2, and RelyX U200 Automix (3M, Maplewood, MN, USA), a dual composite cement, which are applied in the adhesive cementation procedure. The composition of the used materials by the producer is presented in Table 1. 

### 2.2. Methods

#### 2.2.1. Sample Preparation

Elite Glass (Zhermack, Rovigo, Italy), a transparent silicone formulation, was used to prepare polymerisation moulds which provided a space for the material. For the purpose of compressive strength, diametral compressive strength, and microhardness testing, the shape of the polymerisation mould was a cylinder sized 3 mm in diameter and 5 mm in height. For the purpose of flexural strength testing, the shape of the mould was a beam sized 2 × 4 mm. The polymerization moulds prepared in this manner were placed on a metal base on celluloid strips. The moulds were filled with a material and covered with a celluloid strip on the top, and the material surface was levelled using a basic microscope slide. Enamel Plus Hri, a composite material, was heated in the Ena-Heat device (Micerium, Avegno, Italy) to the temperature of 50 °C before polymerisation. The material was polymerised with light generated by Bluephase G2 (Ivoclar Vivadent, Schaan, Lichtenstein), a polymerisation lamp, for 20 s while maintaining direct contact between the fibre optical cable with the surface of the celluloid strip. A Coltolux Light Meter (Sonel, Świdnica, Poland) was used for determining the polymerisation lamp’s light intensity, which was 1200 mW/cm^2^ when the fibre optical cable was in direct contact with the surface of the celluloid strip covering the material. 80 samples were made in total. The test was conducted after 24 h of keeping samples in distilled water. 

#### 2.2.2. Microhardness Measurement

Microhardness was tested in accordance with the Vickers method using a Shimadzu type M (Shimadzu Corporation, Kyoto, Japan), a microhardness tester; the load was 50 g. Microhardness was calculated based on the following Equation (1): (1)$$HV=1854.4\frac{P}{d^{2}}$$
where
*P* is the applied load [g] *d* is the diagonal of the impression [µm].

#### 2.2.3. Compressive Strength and Elastic Modulus Measurement 

The mechanical strength, i.e., compressive strength, test was carried out with the use of an Instron 8501 (Instron, Norwood, MA, USA), a universal strength testing machine, by means of the diametral compressive strength test (Figure 1). The actuator travel speed was 0.5 mm/min. The value of the maximum stress, measured in MPa, which caused the material to crack was calculated and recorded automatically by a computer interfaced with the testing machine. 20 measurements were taken. Compressive strength is calculated based on the following Equation (2): (2)$$Rc=\frac{F}{A}$$
where
*F* is the highest applied load recorded during sample compression [N]*A* is the initial cross-sectional area of a sample [mm^2^].

**Figure 1 polymers-14-02686-f001:**
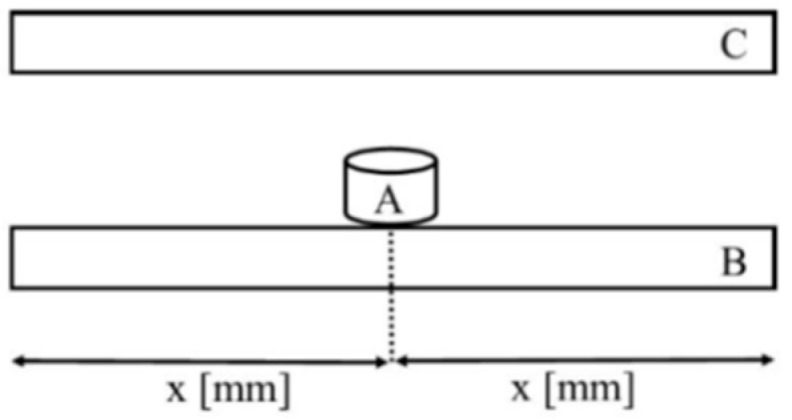
Compressive strength test diagram. Sample (**A**) is placed in the centre of the lower compression anvil (**B**), which is positioned in parallel to the upper compression anvil (**C**).

Based on the data obtained during strength tests, i.e., the compression test, the bulk modulus was calculated. The elastic modulus was defined as stress increment for the relative shortening of a sample’s height within the range from 0.5 to 1.5%. It is determined based on the following correlation:(3)$$E=\frac{\Delta \sigma }{\Delta e}=\frac{\left(\sigma_{1.5}-\sigma_{0.5}\right)}{\varepsilon_{1.5}-\varepsilon_{0.5}}$$
where
Δσ is the stress increment for the assumed range of the relative shortening of a sample,$\Delta \varepsilon =\frac{\Delta h}{h0}$ is the relative change of a sample’s height

#### 2.2.4. Flexural Strength Measurements

A beam-shaped sample of the examined material is placed at an equal spacing on two supports. The third point is the load applied symmetrically in relation to the supports (Figure 2). The load is moved with the appropriate speed (0.75 ± 0.25 mm·min^−1^ as per ISO 4049) perpendicularly to the longitudinal axis of the sample. As the material bends, the upper compressive stress and the lower tensile stress are compensated in the central area. Flexural strength is calculated based on the following Equation (4):(4)$$\sigma =\frac{3\times F\times l}{2\times b\times h^{2}}$$
where
*F* is the maximum applied load at the time of sample destruction [N],*L* is the distance between supports [mm],*B* is the width of the cross-section of a sample [mm],*h* is the height of the cross-section of a sample [mm].

**Figure 2 polymers-14-02686-f002:**
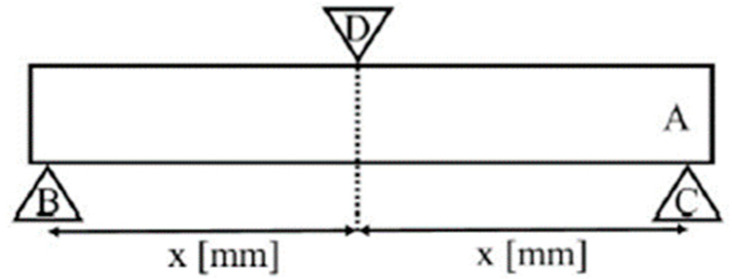
Flexural strength test diagram. Sample (**A**) is placed at an equal spacing on two supports (**B**,**C**). The third point is the load (**D**).

#### 2.2.5. Diametral Tensile Strength

A cylindrical sample of the examined material is placed in the central part of the bottom cross-head positioned in parallel to the top cross-head. The cross-heads are positioned perpendicularly to the direction of the applied force and the top cross-head moves with a specific speed while compressing a sample of the tested material along its diameter. The actuator travel speed was 0.5 mm/min. The compressive force gives rise to tensile stresses in the central part of the sample (Figure 3). Diametral tensile strength is calculated based on the following Equation (5):(5)$$\mathrm{DTS}=\frac{2\cdot F}{\pi \cdot h\cdot d}$$
where
*F* is the highest applied load recorded during sample compression [N], *d* is the diameter of the examined sample [mm],*h* is the thickness of the examined sample [mm].

**Figure 3 polymers-14-02686-f003:**
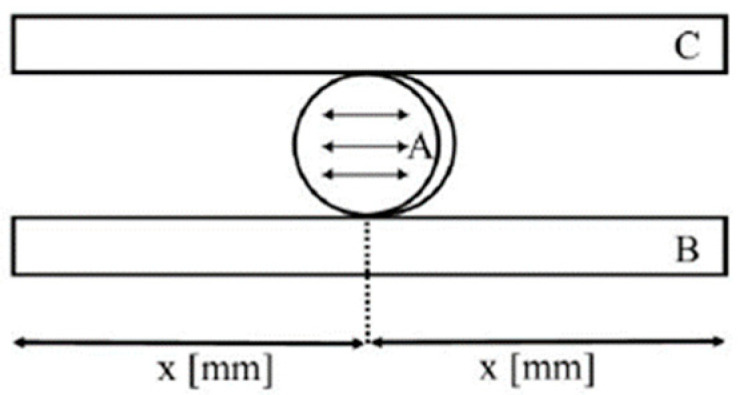
Diametral compressive strength test diagram. Sample (**A**) is placed in the central part of the lower compression anvil (**B**) positioned in parallel to the upper compression anvil (**C**).

#### 2.2.6. Statistical Analysis

Examination was performed on 10 samples for each group of material. The data were evaluated for normal distribution using the Shapiro–Wilk assay. Then, after checking the homogeneity of variance (Brown–Forsythe assay) one test, either a Student’s t-test for independent samples or a Cochran–Cox test with a separate variance estimate, was performed. The analysis was performed with the significance level of *p* ≤ 0.05. All data were computed using the Statistica 13.3 software (StatSoft, Inc., Tulsa, OK, USA). 

## 3. Results

### 3.1. Microhardness

The study of microhardness of the RelyX samples showed the HV parameter of 61.28 MPa (SD = 2.34 MPa), while for the Enamel Plus material HV was 102.56 MPa (SD = 3.52 MPa). Normal distributions, confirmed by the Shapiro–Wilk test, were obtained for both RelyX and Enamel Plus groups (W = 0.9044, *p* > 0.2 and W = 0.9644, *p* > 0.8, respectively) and homogeneity of variance was found using the Brown–Forsythe test (F = 0.803345, *p* = 0.381926). Thus, the conditions for the use of the parametric Student’s *t*-test for independent samples (t = −30.9272, *p* ≤ 0.001) were met. The survey results revealed statistically significant differences between the RelyX and Enamel Plus groups. The value of microhardness of the Enamel Plus material was 67.36% higher than that of RelyX.

### 3.2. Compressive Strength

The study of compressive strength of the RelyX samples showed the CS parameter of 286.4 MPa (SD = 49.7 MPa), while for the Enamel Plus material CS was 406.18 MPa (SD = 121.82 MPa). Normal distributions, confirmed by the Shapiro–Wilk test, were obtained for both RelyX and Enamel Plus groups (W = 0.87139, *p* = 0.10375 and W = 0.86963, *p* = 0.7667, respectively). Due to the lack of homogeneity of variance found using the Brown–Forsythe test (F = 10.03431, *p* = 0.005068), the Cochran–Cox test with a separate variance estimate was performed (t = −2.998281, *p* = 0.009917). The analysis was performed with the significance level of *p* ≤ 0.05. The survey results revealed statistically significant differences between the RelyX and Enamel Plus groups. The value of compressive strength of the Enamel Plus material was 41.84% higher than that of RelyX. 

### 3.3. Elastic Modulus

The study of the elastic modulus of the RelyX samples showed the E parameter of 15.99 GPa (SD = 3.05 GPa), while for the Enamel Plus material E was 18.75 GPa (SD = 2.05 GPa). Normal distributions, confirmed by the Shapiro–Wilk test, were obtained for both the RelyX and Enamel Plus groups (W = 0.9342, *p* > 0.49 and W = 0.9283, *p* > 0.43, respectively). Homogeneity of variance was found using the Brown–Forsythe test (F = 0.1390 *p* = 0.7135) and a Student’s t-test with a separate variance estimate test was performed (t = −2.3723, *p* = 0.03). The analysis was performed with the significance level of *p* ≤ 0.05. The value of the elastic modulus of the Enamel Plus material was 17.75% higher than that of RelyX. The survey results revealed statistically significant differences between the RelyX and Enamel Plus groups. 

### 3.4. Flexural Strength

The study of the flexural strength of the RelyX samples showed the FS parameter of 81.29 MPa (SD = 3.09 MPa), while for the Enamel Plus material FS was 130.72 MPa (SD = 5.99 MPa). Normal distributions, confirmed by the Shapiro–Wilk test, were obtained for both the RelyX and Enamel Plus groups (W = 0.9295, *p* > 0.44 and W = 0.9387, *p* > 0.54, respectively). Due to the lack of homogeneity of variance found using the Brown–Forsythe test (F = 5.9102 *p* = 0.0257), a Cochran–Cox test with a separate variance estimate was performed (t = −22.9444, *p* ≤ 0.001). The analysis was performed with the significance level of *p* ≤ 0.05. The value of flexural strength of the Enamel Plus material was 36.03% higher than that of RelyX. The survey results revealed statistically significant differences between the RelyX and Enamel Plus groups. 

### 3.5. Diametral Tensile Strength

The study of the diametral tensile strength of the RelyX samples showed the DTS parameter of 37.36 MPa (SD = 1.60 MPa), while for the Enamel Plus material DTS was 46.52 MPa (SD = 1.91 MPa). Normal distributions, confirmed by the Shapiro–Wilk test, were obtained for both the RelyX and Enamel Plus groups (W = 0.9780, *p* > 0.95 and W = 0.9414, *p* > 0.57, respectively). Homogeneity of variance was found using the Brown–Forsythe test (F = 0.0781 *p* = 0.7830). A Student’s *t*-test with a separate variance estimate test was performed (t = −11.6270, *p* ≤ 0.001). The analysis was performed with the significance level of *p* ≤ 0.05. The value of diametral tensile strength of the Enamel Plus material was 45.52% higher than that of RelyX. The survey results revealed statistically significant differences between the RelyX and Enamel Plus groups. 

## 4. Discussion

Composite materials are used in more and more applications, not only in dentistry but also in other fields of medicine [10,11]. Examining mechanical properties of dental materials permits a better determination of a given material’s capacity to resist masticatory forces and damage resistance. This is, to a large extent, decisive for the durability of the restorations applied in the oral cavity environment. Enamel Plus Hri, a nanohybrid composite material, was selected for testing, since Micerdent was one of the first companies to develop a compatible system to work with a heated composite, which enabled strict compliance with the manufacturer’s recommendations. RelyX U200 Automix (3M, Maplewood, MN, USA), a dual self-adhesive system with a considerably simplified clinical procedure that mitigated the risk of error during adhesive cementation of prosthetic restorations, was chosen from among a large group of composite cements.

Hardness determines the material’s resistance to the indenter being pushed into its surface vertically with an appropriate load, which results in a permanent deformation of the sample [12]. In order to properly assess the hardness of composites or cements containing fillers with significantly smaller particle sizes, the nanoindentation method is employed, since it uses smaller indenter sizes and a smaller load [13]. Microhardness depends on the composition of the material, in particular, the type of the applied monomers, the content of the filler, as well as the shape and size of filler particles, as well as the radiation energy density, and, what follows, the degree of conversion. Microhardness is an exponent of the material’s resistance to mechanical damage. Decreased microhardness might cause mechanical damage to the material placed between the tooth substance and the ceramic restoration, which might result in a marginal gap, development of caries, and decementation of the restoration. The microhardness measurement is an indirect method of assessing the degree of conversion of carbon double bonds in resin composites. Therefore, the greater the microhardness, the smaller the polymerisation shrinkage and, consequently, the lower the risk of marginal leakage in the tooth–cement and cement–indirect restoration configurations [14]. The degree of conversion is a measure of progress of the polymerisation reaction and, in the case of methacrylate resin curing, it determines the percentage ratio of the number of double methacrylate bonds which have undergone reactions and their initial content in the monomer. The progress of the photopolymerisation reaction, and hence the value of the degree of conversion (DC) parameter, depends on many factors, such as the composition of the cured material, geometry of the sample exposed to light, concentration of photoinitiators, intensity of the light cast on the sample, light exposure time, temperature of the polymerisation process, access of oxygen to the reaction environment, and the light exposure method (with a constant or a variable intensity) [7,15]. Heating the composite material, as opposed to leaving the material at room temperature, results in an increased microhardness and degree of conversion. This is an effect of the reduced viscosity of the material following preheating, of the higher mobility of free radicals, and of the increased frequency of collisions of non-reactive groups. In their meta-analysis, Elkaffas compared thirteen papers discussing microhardness of a heated composite material at various temperatures. Based on the analysis, they concluded that heating a material increases microhardness, which improves the mechanical properties of the material [16]. The tests carried out in the abovementioned study showed higher values of microhardness for a heated composite material compared to a composite cement at the room temperature. As already mentioned, microhardness depends on the filler structure: quantity, particle size, and shape. This thesis is confirmed by the studies cited below and by the results achieved in this article. Kashi carried out a microhardness test of three materials based on a nanocomposite filler by the Vickers method. The materials were tested at room temperature, at 37 °C and at 54 °C. Regardless of temperature, the highest values were achieved by the composite material having the highest filler content, Grandio, with the average microhardness value being 94 HV (71.4% filler content) [14]. Ayub tested three microhybrid materials with varying contents of the filler and one material with a nanofiller. The highest microhardness values were achieved by the preheated nanocomposite material, Filtek Supreme Ultra, characterised by the lowest size of the filler particles and the highest percentage content of the filler [17]. The above theses are confirmed by our own research. The microhardness value decrease between the tested materials was 67.36%, which should be explained mainly by the increased filler content in the Enamel Plus HRi relative to the RelyX U200 Automix cement (63 and 43%, respectively). The authors cannot refer directly to other publications because there is no scientific research comparing microhardness of a heated composite material and a composite cement. Due to the fact that the microhardness of a heated composite material was 67.36% higher than that of a composite cement, an increased resistance to mechanical damage, a higher degree of conversion, and a smaller marginal leakage of the heated composite material are expected. In order to confirm the thesis stated above, a range of subsequent tests should be conducted with a focus on evaluating the tightness of the connection in the tooth–cement and cement–indirect restoration configurations by means of a scanning electron microscope (SEM).

Tooth structure stresses during the application of forces are compound and can be broken down into the following basic types: compression, tension, and shear [18]. This is critical for the durability of cement due to the high values of the forces in the masticatory system, such as 100–150 N for molar teeth, occurring while food is crushed [19]. Compressive strength, which is the resistance against the masticatory force, is an important factor of clinical performance. Compressive strength is defined as the maximum stress recorded during sample compression. The compression test allows the determination of the elastic modulus, proof stress, deformation after exceeding yield point, and compressive strength. The compression test identifies how materials behave under crushing forces. A material’s compressive strength is influenced most by the type and content of the filler and the composition of the organic matrix [20,21]. A greater percentage content of the filler increases the material’s mechanical strength and, among others, compressive strength under crushing forces. This is confirmed by the results obtained in the study mentioned above: The CS of the RelyX U200 self-adhesive cement with the filler content of 43 vol% is 286.36 MPa compared to that of the heated Enamel Plus HRi material with the filler content of 63% is 406.18 MPa. Similar observations were made by Nada et al., who examined compressive strength of three different composite materials (Clearfil Majesty, Z-100, Light-Core). Each of the examined materials was characterised by a different filler content (92%, 85%, 80.5%). The highest compressive strength was achieved by the composite material heated to a temperature of 52 °C and with the highest content of the Clearfil Majesty filler, i.e., 369.75 MPa [22]. Ah-Rang, in turn, examined the mechanical properties of self-adhesive composite cements using different activation modes (self-cured, light-cured) and testing time (immediately, 24 h, and thermocycling). The results concerning the RelyX U200 cement are very similar to the authors’ findings achieved in their own research. The light-activated RelyX U200 cement was tested after 24 h with a compressive strength of 231.91 MPa [23]. The authors of the above studies, in turn, obtained the result of 286.36 MPaZ in the same conditions. Enamel Plus’s compressive strength being higher than that of RelyX by 41.84% determines a greater resistance to compressive forces, which play an important role in the mastication process [24]. 

Diametral tensile strength (DTS) enables the determination of a brittle dental material’s capacity to resist the tensile stresses occurring during mastication. DTS depends on the type of the material, the composition of the organic matrix, the characteristics of the filler, the bond between the filler and the matrix, and the conditions of polymerisation. The higher the DTS value, the more resistant the material to compound forces acting in the oral cavity during mastication [5]. Sokołowska examined the DTS of the Enamel Plus Hri composite material heated to a temperature of 39 °C and 50 °C compared to the room temperature of 20 °C, i.e., the same material used by the authors in their own research. Sokołowska obtained 55.7 MPa for the material heated to the temperature of 50 °C, while the authors own research obtained the value of 46.52 MPa in the same conditions [25]. Sokołowski, in turn, examined the DTS of various composite cements, e.g., the RelyX U200 cement, in their study [26]. The obtained diametral compressive strength values of 38 MPa are very similar to the values obtained in the authors’ own research regarding the same material. The authors obtained the DTS result of 37.36%. In the presented tests, Enamel Plus achieved a 45.52% higher result than RelyX 200 Automix cement, which should be explained by the varying filler contents in both materials. Moreover, an addition of functional particles responsible for the adhesion process in RelyX cement may weaken the connection between the filler and the polymer matrix, thereby worsening the material’s mechanical properties. Similar observations were made by Oguri, who proved that an addition of functional monomers, MAC-10, responsible for the adhesion process decreases the conversion process and hence explains the low DTS values [27]. 

Flexural strength, in turn, is the maximum stress recorded during sample bending at the time of its destruction. Flexural strength depends mainly on the composition of the organic matrix and the characteristics of the filler (type, content, morphology) [28]. The authors of our own research obtained higher FS values: 130.72 MPa for the heated composite material, where the percentage filler volume was 63%, and for the RelyX cement it was 81.29MPa with the filler volume being 43%. The above thesis that FS depends primarily on the characteristics (quantity) of the filler is confirmed by numerous studies by other authors, who indicate also that the heating alone of a material does not have a considerable influence on flexural strength [29,30,31]. D’Amario made similar observations. They tested three different composite materials at room temperature and then heated to 39 °C. The highest result was achieved by Opallis (120 MPa), which was characterised by the highest filler content: 57% [32]. In their subsequent studies, these authors also observed a strong correlation between bending strength and filler content in the material. Three composite materials were tested: Enamel Plus HFO, Enamel Plus HRi, and Opallis plus. The highest result was achieved by Enamel Plus HRi: 128 MPa, with 63% filler content [33]. Hence, it can be concluded that the increased filler content in a material results in a higher flexural strength and, what follows, a higher mechanical strength to the crushing forces occurring during food mastication. 

The elastic modulus determines the degree of elasticity of materials, which is attributable to the forces occurring between atoms or molecules of a material. As the primary forces of attraction increase, the value of the elastic modulus grows, which makes the material more stiff [34]. The values of the elastic modulus of dental materials are strongly correlated with the elastic modulus of enamel and dentine. The elastic modulus of enamel is approx. 84 GPa depending on its chemical structure and spatial distribution of prisms, which was shown by the calculations performed in accordance with the finite element method [11]. In their studies, Watanabe draw the conclusion that the elastic modulus of dentine depends on whether it is mineralised or demineralised dentine. According to the authors’ calculations, the elastic modulus of mineralised dentine is 17.4 GPa, while that of demineralised dentine drops to as low as 1.4 GPa [35]. The material used for reconstructing and cementing a prosthetic restoration should handle the stresses occurring during mastication and, therefore, the more similar the strength parameters of a material to those of the natural tooth substances, mainly dentine, the better its function in the stomatognathic system [36]. The findings of the authors of the above studies demonstrate that the elastic modulus of both a self-adhesive cement (15.99 GPa) and a heated composite material (18.76 GPa) is similar to that of dentine, which is a good characteristic of both materials.

It needs to be emphasised that there is no scientific research comparing the mechanical properties of a heated composite material and an adhesive cement. This topic seems to be significant in the clinical practice of a dentist. The methods of laboratory measurement of material strength enable a comparison of the physicomechanical properties forecasting their preservation under occlusal load. 

## 5. Conclusions

1. Due to its better mechanical properties, heated composite material is an advantageous alternative to composite cements in the indirect restoration placement procedure. 

2. As the strength parameters of the heated composite material increase, a higher resistance to the compressive and bending forces present in the oral cavity, and hence a greater durability of the created reconstructions, can be expected.

3. Due to higher values of the microhardness parameter and, indirectly, the degree of conversion, a lower risk of marginal leakages, and consequently of development of secondary caries between the tooth substance and the indirect restoration, is expected.

## Figures and Tables

**Table 1 polymers-14-02686-t001:** Chemical composition of the tested materials.

Material	Classification	Composition	Total Content of Filler
Enamel Plus Hri	Microhybrid composite material	UDMA, Bis-GMA, 1,4-butandioldimethacrylate; Glass filler, highly dispersed, silicone dioxide	80% by weight 63% by volume
Rely X U200	Dual-cured self-adhesive composite cement	Bis-phenol-A-bis-(2-hydroxy-3-methacryloxypropyl)ether (Bis-GMA), Triethylenglycodimethacrylate (TEGDMA) Silanized glass and silica filler; Initiator system: sodium p-toluen sulfinate, camphorquinone;	43% by volume

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
