# Peer review of "Comparison of Mechanical Properties of a Self-Adhesive Composite Cement and a Heated Composite Material"

_polymers, 2022, doi:10.3390/polym14132686_

Round 1

Reviewer 1 Report

Comments 1: In this work, the authors choose two composite materials (one was self-adhesive composite cement, the other was heated composite material), which were applied in adhesive cementation procedure of dental practices, to compared the mechanical properties of these two materials. From the results of comparison, including microhardness, compressive strength, flexural strength, diametral compressive strengthen, and elastic modulus, the authors draw the conclusion of that the heated composite material with better mechanical properties is a promising alternative in the indirect restoration placement procedure.

Although, comprehensive experiments had been conducted and came up with consistent results before the authors giving the conclusion, there are many aspects to be improved.

Comments 2: For 'Abstract', authors just declared the main aim of this study, as 'what we want to do', in 'background' section. But this should be presented in 'Method' section rather than 'background'. In 'background' section, authors should present the current state of the field and the main problems/limits that urgent to be solved.

Comments 3: For 'Introduction', there was similar problem as 'background' section of 'Abstract'. Authors just listed the composition, applying principle, limits, and current applications of materials for hard tooth substance preparations that studied in the work. But there was no good logical organization for the background information, so that the readers cannot clearly comprehend the highlights, significance, and the main contribution for research field of this study. In addition, there are too many unnecessary description and inserts that make the introduce statement lengthy and hazy for comprehension. Please try to keep scientific and concise description always, if a phrase or short sentence is deleted without affecting the comprehension of the whole sentence, then please delete it.

For instance:

The 'increasingly' in the statement of 'They are also used increasingly more frequently instead of large direct restorations' at row 34 has similar mean of 'more';

The insert of 'which, given correct preparation' at row 77 is unnecessary description.

Comments 4: For 'Results', the statistical definitions such as 'The analysis was performed for the significance level of p ≤ 0.05' at row 193 to 194, should be detailed in the 'Method' section rather than 'Results'.

Comments 5: For 'Discussion', there was too much background introduction about the materials and mechanical parameters, and there was lack of scientific analysis and systematic comparison of the results draw out from this work to outstand the highlights and significance of this study. Recent advances on adhesive composites (VIEW 2021, 2:20200165; VIEW 2021, 2:20200081; Small Methods 2021, 5, 2001250) and biomedical applications (Angewandte Chemie International Edition, 2020, 59(27): 10831; Advanced NanoBiomed Research 2021, 1, 2000104; Matter 2019, 1, 1669-1680) should be included and discussed.

Comments 6: The figures and figure captions in this manuscript was too rough to lead the reader to understand the content of the study.

For instance:

There was only one sentence of the figure captions in Figure 1 to 3, and did not explain the symbols such as 'x', 'A', 'B', and 'C' in the figures.

The box plots of Figure 4 to 8 were lack of displaying corresponding data dots and necessary legends.

The original legend was not deleted in Figure 6.

Comments 7: There were some mistakes in abbreviation formats. All abbreviations should be fully written, when it was firstly mentioned, and the corresponding abbreviations should be written in the brackets afterwards.

For instance:

DEGMA (diethylene glycol dimethacrylate) at row 48;

TEGDMA (triethylene glycol dimethacrylate) at row 48;

Bis-GMA (bisphenol A-glycidyl methacrylate) at row 50.

Comments 8:

There were many mistakes in the text, such as extra characters and space bars, and the space bars were missed in some other places.

For instance:

The extra ';];' at row 59;

The extra space bars at row 22 and 137;

Two space bars of 'p>0.2' at row 190 were missed, it should be 'p > 0.2'. And there were many other places missed space bars like this.

Comments 9:

It is necessary to polish the language and gramma by other native English speakers.

Author Response

Dear Reviewer,

Thank you very much for the perceptive review, constructive comments on our manuscript and the time and energy devoted to help us improve the paper. According to your implications and comments we have rearranged and improved the entire text of the manuscript with a special emphasis on the excerpts identified in the review.

Point 1:  In this work, the authors choose two composite materials (one was self-adhesive composite cement, the other was heated composite material), which were applied in adhesive cementation procedure of dental practices, to compared the mechanical properties of these two materials. From the results of comparison, including microhardness, compressive strength, flexural strength, diametral compressive strengthen, and elastic modulus, the authors draw the conclusion of that the heated composite material with better mechanical properties is a promising alternative in the indirect restoration placement procedure.

Although, comprehensive experiments had been conducted and came up with consistent results before the authors giving the conclusion, there are many aspects to be improved.

Point 2: For 'Abstract', authors just declared the main aim of this study, as 'what we want to do', in 'background' section. But this should be presented in 'Method' section rather than 'background'. In 'background' section, authors should present the current state of the field and the main problems/limits that urgent to be solved.

Response 2: Thank you for the suggestion, I’ll make changes to the abstract.

Point 3: For 'Introduction', there was similar problem as 'background' section of 'Abstract'. Authors just listed the composition, applying principle, limits, and current applications of materials for hard tooth substance preparations that studied in the work. But there was no good logical organization for the background information, so that the readers cannot clearly comprehend the highlights, significance, and the main contribution for research field of this study. In addition, there are too many unnecessary description and inserts that make the introduce statement lengthy and hazy for comprehension. Please try to keep scientific and concise description always, if a phrase or short sentence is deleted without affecting the comprehension of the whole sentence, then please delete it.

For instance:

The 'increasingly' in the statement of 'They are also used increasingly more frequently instead of large direct restorations' at row 34 has similar mean of 'more';

The insert of 'which, given correct preparation' at row 77 is unnecessary description.

Response 3: Thank you for the suggestion, I’ll make changes

Point 4:  For 'Results', the statistical definitions such as 'The analysis was performed for the significance level of p ≤ 0.05' at row 193 to 194, should be detailed in the 'Method' section rather than 'Results'.

Response 4: Thank you for the suggestion, I’ll make changes

Point 5: For 'Discussion', there was too much background introduction about the materials and mechanical parameters, and there was lack of scientific analysis and systematic comparison of the results draw out from this work to outstand the highlights and significance of this study. Recent advances on adhesive composites (VIEW 2021, 2:20200165; VIEW 2021, 2:20200081; Small Methods 2021, 5, 2001250) and biomedical applications (Angewandte Chemie International Edition, 2020, 59(27): 10831; Advanced NanoBiomed Research 2021, 1, 2000104; Matter 2019, 1, 1669-1680) should be included and discussed.

Response 5: Thank you for the suggestion, I’ll make changes

Point 6: he figures and figure captions in this manuscript was too rough to lead the reader to understand the content of the study.

For instance:

There was only one sentence of the figure captions in Figure 1 to 3, and did not explain the symbols such as 'x', 'A', 'B', and 'C' in the figures.

The box plots of Figure 4 to 8 were lack of displaying corresponding data dots and necessary legends.

The original legend was not deleted in Figure 6.

Response 6: Thank you for the suggestion, I’ll make changes

Point 7: There were some mistakes in abbreviation formats. All abbreviations should be fully written, when it was firstly mentioned, and the corresponding abbreviations should be written in the brackets afterwards.

For instance:

DEGMA (diethylene glycol dimethacrylate) at row 48;

TEGDMA (triethylene glycol dimethacrylate) at row 48;

Bis-GMA (bisphenol A-glycidyl methacrylate) at row 50.

Response 7: Thank you for the suggestion, I’ll make changes

Reviewer 2 Report

Dear authors,

on the end of the introduction part you should write the aim of your study. I suggest to delete Table 1. The section Materials and Methods, as well as Results should be written in past tense. Methods are written on the way which is very hard to understand, please clarify. Delete Figure 4., there is no need to have a graphical part if you already mentioned the results in the text. The same for the Figures 5-8. Correct reference number 30.

I strongly suggest proof-reading for your manuscript.

Author Response

Dear Reviewer,

Thank you very much for the perceptive review, constructive comments on our manuscript and the time and energy devoted to help us improve the paper. According to your implications and comments we have rearranged and improved the entire text of the manuscript with a special emphasis on the excerpts identified in the review.

Responses to the Reviewer  comments:

on the end of the introduction part you should write the aim of your study. I suggest to delete Table 1. The section Materials and Methods, as well as Results should be written in past tense. Methods are written on the way which is very hard to understand, please clarify. Delete Figure 4., there is no need to have a graphical part if you already mentioned the results in the text. The same for the Figures 5-8. Correct reference number 30.

I strongly suggest proof-reading for your manuscript.

REPLY: Thank you very much for all suggestions. I have added the aim of the study at the end of the introduction. Table 1 presents the composition of the tested materials, which is important for understanding why a given material is characterised by lower/higher values of a given mechanical property. Figures 4–8 present a graphical distribution of the results, which can be more transparent to the reader. Language mistakes will be corrected.

We hope that the manuscript so amended in line with the suggestions made by the Interviewers will prove to be satisfactory and will meet the criteria required for its publication.

Yours sincerely,

Authors

Reviewer 3 Report

The authors compared the mechanical properties of RelyX and Enamel 216 Plus materials. This manuscript looks like a technical report instead of a scientific article. Some comments to be addressed are listed as follows:

1.    Line 111, ‘The test was conducted after 24 hours of keeping samples in distilled water.’ Water absorption has been widely reported to influence the mechanical properties of composite material. The reviewer would like to know how the authors tackled this issue.

2.    For the compression test, Line 124, The actuator travel speed was 0.5 mm/min. For the flexural strength measurements, Line 149, The load is moved with the appropriate speed (0.75 ± 0.25 mm/min). For the diametral tensile strength measurements, the loading speed is not given. In addition, how to evaluate the loading rate effect on these measurements?

3. Fig.1, Compressive strength test diagram, the two corners at the bottom of the cylindrical specimen is not in contact with the loading platen. The end effect and stress concentration are unclear.

4. Line 136, ‘Based on the data obtained during strength tests, i.e. the compression test, the bulk modulus was calculated. The elastic modulus was defined as stress increment for the relative shortening of a sample’s height within the range from 0.5 to 1.5%.’ Pls introduce how the sample’s height change is measured. Is it based on the total (compliance) displacement from the Instron, or the direct measurement of the specimen through the real time image analysis? If it is only from the Instron machine, the measured relative change of sample’ height would be not very accurate, and consequently the modulus.

5. Section 3.4 and 3.5, why the flexural strength is much higher than the diametral tensile strength measured from the Brazilian disk? How about the stress concentration at two ends of the Brazilian disk? This results in a lower diametral tensile strength.

6. The mechanisms in the different mechanical properties of RelyX and Enamel 216 Plus materials are unclear.

7. In the discussion, pls compare the current results with the available data in the literature.

Author Response

Dear Reviewer,

Thank you very much for the perceptive review, constructive comments on our manuscript and the time and energy devoted to help us improve the paper. According to your implications and comments we have rearranged and improved the entire text of the manuscript with a special emphasis on the excerpts identified in the review.

Responses to the Reviewer comments:

  • Line 111, ‘The test was conducted after 24 hours of keeping samples in distilled water.’ Water absorption has been widely reported to influence the mechanical properties of composite material. The reviewer would like to know how the authors tackled this issue.

The primary environment where composite materials and cements interact after application is the oral cavity. The oral cavity is moistened with saliva in the physiological conditions. Saliva is composed in 99% of water and organic and inorganic components. Sample storage in distilled water reflects the conditions prevailing in the oral cavity to the best extent. In addition, research points that storage in water causes the lowest degradation of the components of the material.

REPLY:

  • For the compression test, Line 124, The actuator travel speed was 0.5 mm/min. For the flexural strength measurements, Line 149, The load is moved with the appropriate speed (0.75 ± 0.25 mm/min). For the diametral tensile strength measurements, the loading speed is not given. In addition, how to evaluate the loading rate effect on these measurements?

REPLY: The speed of the DTS test, similarly to the diametral compressive strength, is set to 0.5 mm/min.

Parameters determining test conditions have not been specified so far by any standard referring to dentistry polymer materials for restorations. A certain suggestion can be found in the conditions determined in PN-EN ISO 604 “Plastics. Determination of compressive properties”, which defines the optimum dimensions of samples for testing, their quantity, and size of initial load.

  • 1, Compressive strength test diagram, the two corners at the bottom of the cylindrical specimen is not in contact with the loading platen. The end effect and stress concentration are unclear.

REPLY: Thank you for the suggestion. I’ll make the changes.

  • Line 136, ‘Based on the data obtained during strength tests, i.e. the compression test, the bulk modulus was calculated. The elastic modulus was defined as stress increment for the relative shortening of a sample’s height within the range from 0.5 to 1.5%.’ Pls introduce how the sample’s height change is measured. Is it based on the total (compliance) displacement from the Instron, or the direct measurement of the specimen through the real time image analysis? If it is only from the Instron machine, the measured relative change of sample’ height would be not very accurate, and consequently the modulus.

REPLY: The change in the sample height was measured with the use of a dynamic extensometer (Instron 2620-601) with the measuring range of +/- 5 mm.  We did not use any of the aforementioned measurement methods in our tests. We used the sample shortening measured with the use of an extensometer. The measurement section corresponded to the sample height.

  • Section 3.4 and 3.5, why the flexural strength is much higher than the diametral tensile strength measured from the Brazilian disk? How about the stress concentration at two ends of the Brazilian disk? This results in a lower diametral tensile strength.

REPLY: The bending strength is approx. 2 x higher for plastic materials because stress gradient occurs on the sample’s cross-section, and compressive stress occurs in the upper part, i.e. located where the load is applied, and tensile stress occurs in the lower part, which is the reason why there is zero stress in the middle part. At the same time, cracks are blocked in the compressed part, which results in a higher strength during bending as compared to tension.

  • The mechanisms in the different mechanical properties of RelyX and Enamel 216 Plus materials are unclear.

REPLY: I have corrected the discussion and described these mechanisms there.

  • In the discussion, pls compare the current results with the available data in the literature.

As mentioned in the text, there are no current studies concerning a comparison of cements and preheated composite materials in a single study. In the discussion, the authors focus on the current studies concerning preheated composite materials and cements.

Round 2

Reviewer 1 Report

The authors should include point-to-point responses in the cover letter, so that reviewers can track the changes clearly.

Also, the references should be updated and key literatures still seem to be missing.

Author Response

Dear Reviewer,

          Thank you very much for the perceptive review, constructive comments on our manuscript and the time and energy devoted to help us improve the paper. I answered the previous review point-by-point and I marked the changes in red in the manuscript. I have added six current and suggested bibliography items (also marked in red). I hope that changes will be clear for you. 

Kind regards

Anastazja Skapska

Reviewer 2 Report

Dear authors,

please delete the unnecessary tables and graphs, please indicate all changes you have made in the manuscript, including proof reading. please add the proof reading certificate

Author Response

Dear Reviewer,

Thank you very much for the perceptive review, constructive comments on our manuscript and the time and energy devoted to help us improve the paper. I delete unnecessary graphs and I marked all changes in red. Also I add proof reading certificate. 

Kind regards

Anastazja SkÄ…pska 

Reviewer 3 Report

The authors have addressed the comments. The quality of the manuscript is improved. This revised version is suggested for publication.  

Author Response

Dear Reviewer, 

Kind regards

Anastazja SkÄ…pska